# A fully automatic AI system for tooth and alveolar bone segmentation from cone-beam CT images

Zhiming Cui [1,2,3,10], Yu Fang [1,10], Lanzhuju Mei [1,10], Bojun Zhang[4,10], Bo Yu[5], Jiameng Liu[1], Caiwen Jiang[1], Yuhang Sun[1], Lei Ma[1], Jiawei Huang[1], Yang Liu[6], Yue Zhao[7✉], Chunfeng Lian[8✉], Zhongxiang Ding[9✉], Min Zhu[4✉] & Dinggang Shen[1,3✉]

Accurate delineation of individual teeth and alveolar bones from dental cone-beam CT (CBCT) images is an essential step in digital dentistry for precision dental healthcare. In this paper, we present an AI system for efficient, precise, and fully automatic segmentation of real-patient CBCT images. Our AI system is evaluated on the largest dataset so far, i.e., using a dataset of 4,215 patients (with 4,938 CBCT scans) from 15 different centers. This fully automatic AI system achieves a segmentation accuracy comparable to experienced radiologists (e.g., 0.5% improvement in terms of average Dice similarity coefficient), while significant improvement in efficiency (i.e., 500 times faster). In addition, it consistently obtains accurate results on the challenging cases with variable dental abnormalities, with the average Dice scores of 91.5% and 93.0% for tooth and alveolar bone segmentation. These results demonstrate its potential as a powerful system to boost clinical workflows of digital dentistry.

[1] School of Biomedical Engineering, ShanghaiTech University, Shanghai 201210, China. [2] Department of Computer Science, The University of Hong Kong, Hong Kong 999077, China. [3] Shanghai United Imaging Intelligence Co., Ltd., Shanghai 200030, China. [4] Shanghai Ninth People's Hospital, Shanghai Jiao Tong University, Shanghai 200011, China. [5] School of Public Health, Hangzhou Medical College, Hangzhou 310013, China. [6] Department of Orthodontics, Stomatological Hospital of Chongqing Medical University, Chongqing 401147, China. [7] School of Communication and Information Engineering, Chongqing University of Posts and Telecommunications, Chongqing 400065, China. [8] School of Mathematics and Statistics, Xi'an Jiaotong University, Xi'an 710049, China. [9] Department of Radiology, Hangzhou First People's Hospital, Zhejiang University, Hangzhou 310006, China. [10]These authors contributed equally: Zhiming Cui, Yu Fang, Lanzhuju Mei, Bojun Zhang. ✉email: zhaoyue@cqupt.edu.cn; chunfeng.lian@xjtu.edu.cn; hangzhoudzx73@126.com; ZHUM1612@sh9hospital.org.cn; dgshen@shanghaitech.edu.cn

With the improved living standards and elevated awareness of dental health, an increasing number of people are seeking dental treatments (e.g., orthodontics, dental implants, and restoration) to ensure normal function and improve facial appearance[1–3]. As reported by the Oral Disease Survey[4], nearly 90% of people in the world suffer from a certain degree of dental problems, and many of them need dental treatments. In clinical practice of dental treatments, medical imaging with different modalities, such as 2D panoramic X-rays, 3D intra-oral scans, and 3D cone-beam computed tomography (CBCT) images, are commonly acquired to assist diagnosis, treatment planning, and surgery. Among all available options, CBCT imaging is a sole modality to provide comprehensive 3D volumetric information of complete teeth and alveolar bones. Hence, segmenting individual teeth and alveolar bony structures from CBCT images to reconstruct a precise 3D model is essential in digital dentistry.

Although automatic segmentation of teeth and alveolar bones has been continuously studied in the medical image computing community, it is still a practically and technically challenging task without any clinically applicable system. Many methods have been explored over the last decade to design hand-crafted features (e.g., level set, graph cut, or template fitting) for tooth segmentation[5–13]. These low-level descriptors/features are sensitive to complicated appearances of dental CBCT images (e.g., limited intensity contrast between teeth and surrounding tissues), thus requiring tedious human interventions for initialization or post-correction. Recently, deep learning, e.g., based on convolutional neural networks (CNNs), shows promising applications in various fields due to its strong ability of learning representative and predictive features in a task-oriented fashion from large-scale data[14–23]. Encouraged by the great success of deep learning in computer vision and medical image computing, a series of studies attempt to implement deep neural networks for tooth and/or bony structure segmentation[24–30]. However, these existing methods are still far from fully automatic or clinically applicable, due to three main challenges. First, fully automatic tooth and alveolar bone segmentation is complex consisting of at least three main steps, including dental region of interest (ROI) localization, tooth segmentation, and alveolar bone segmentation. Previous works cannot conduct all these steps fully automatically in an end-to-end fashion, as they typically focus only on a single step, such as tooth segmentation on predefined ROI region[24–30] or alveolar bone segmentation[31,32]. Second, it is hard to handle complicated cases commonly existing in clinical practice, e.g., CBCT images with dramatic-variations in structures scanned from patients with dental problems (e.g., missing teeth, misalignment, and metal artifacts). Third, previous methods are usually implemented and tested on very small-sized datasets (i.e., 10–30 CBCT scans), limiting their generalizability or applicability on the CBCT images acquired with different imaging protocols and diverse patient populations.

In this study, we develop a deep-learning-based AI system that is clinically stable and accurate for fully automatic tooth and alveolar bone segmentation from dental CBCT images. In particular, for tooth segmentation, an ROI generation network first localizes the foreground region of the upper and lower jaws to reduce computational costs in performing segmentation on high-resolution 3D CBCT images. Then, a specific two-stage deep network explicitly leverages the comprehensive geometric information (naturally inherent from hierarchical morphological components of teeth) to precisely delineate individual teeth. Concurrently, for alveolar bone segmentation, a specific filter-enhanced network first enhances intensity contrasts around bone boundaries and then combines the enhanced image with the original one to precisely annotate bony structures. To validate the robustness and generalizability of our AI system, we evaluate it on the largest dataset so far (i.e., 4938 CBCT scans of 4215 patients) from 15 different centers with varying data distributions. In addition, the clinical utility or applicability of our AI system is also carefully verified by a detailed comparison of its segmentation accuracy and efficiency with two expert radiologists.

## Results

**Study design and participants.** In this work, we collected large-scale CBCT imaging data from multiple hospitals in China, including the Stomatological Hospital of Chongqing Medical University (CQ-hospital), the First People's Hospital of Hangzhou (HZ-hospital), the Ninth People's Hospital of Shanghai Jiao Tong University (SH-hospital), and 12 dental clinics. All dental CBCT images were scanned from patients in routine clinical care. Most of these patients need dental treatments, such as orthodontics, dental implants, and restoration. In total, we collected 4938 CBCT scans of 4215 patients (mean age: 38.4, with 2312 females and 1903 males) from the CQ-hospital, HZ-hospital, and SH-hospital as the internal dataset, and 407 CBCT scans of 404 patients from the remaining 12 dental clinics as the external dataset.

The detailed imaging protocols of the studied data (i.e., image resolution, manufacturer, manufacturer's model, and radiation dose information of tube current and tube voltage) are listed in Table 1. To intuitively show the image style variations across different manufacturers caused by radiation dose factors (i.e., tube current, tube voltage, etc), we also provide a heterogeneous intensity histogram of the CBCT data collected from different centers and different manufacturers. As shown in Fig. 1a, we can find that there are large appearance variations across data, indicating necessity of collecting a large-scale dataset for developing an AI system with good robustness and generalizability. Besides the demographic variables and imaging protocols, Table 1 also shows data distribution for dental abnormality, including missing teeth, misalignment, and metal artifacts. Notably, as a strong indicator of clinical applicability, it is crucial to verify the feasibility and robustness of an AI-based segmentation system on challenging cases with dental abnormalities as commonly encountered in practice. To define the ground-truth labels of individual teeth and alveolar bones for model training and performance evaluation, each CBCT scan was manually annotated and checked by senior raters with rich experience (see details in Supplementary Fig. 1).

As shown in Fig. 1b, in our experiments, we randomly sampled 70% (i.e., 3172) of the CBCT scans from the internal dataset (CQ-hospital, HZ-hospital, and SH-hospital) for model training and validation; the remaining 30% data (i.e., 1359 scans) were used as the internal testing set. Moreover, to further evaluate how the learned deep learning models can generalize to the data from completely unseen centers and patient cohorts, we used the external dataset collected from 12 dental clinics for independent testing. To verify the clinical applicability of our AI system in more detail, we randomly selected 100 CBCT scans from the external set, and compared the segmentation results produced by our AI system and expert radiologists. Moreover, we also provide the data distribution of the abnormalities in the training and testing dataset. As shown in Supplementary Table 1 in Supplementary Materials, we can see that the internal testing set and the training set have similar distributions of dental abnormalities, as they are randomly sampled from the same large-scale dataset. In contrast, since the external dataset is collected from different dental clinics, the distribution of its dental abnormalities is a little different compared with the internal set. Notably, some subjects may simultaneously have

**Table 1 Description and characteristics of CBCT dataset from different centers, including internal set (for training, and testing) and external testing set.**

| Cohorts | | Internal set | | | External testing set |
|---|---|---|---|---|---|
| | | CQ-Hospital | HZ-Hospital | SH-Hospital | |
| Demographic variables | CBCT number | 1532 | 1798 | 1201 | 407 |
| | Patient number | 924 | 1689 | 1198 | 404 |
| | Female | 516 | 970 | 547 | 279 |
| | Male | 408 | 719 | 651 | 125 |
| | Age (years) | 40.6 (6, 84) | 37.9 (6, 86) | 38.9 (4, 90) | 29.5 (6–85) |
| Imaging protocols | Resolution (mm) | 0.40 (1473 cases) | 0.40 (1194 cases) | 0.30 (196 cases) | 0.50 (30 cases) |
| | | | | 0.20 (3 cases) | 0.38 (51 cases) |
| | | 0.30 (43 cases) | 0.20 (601 cases) | 0.16 (997 cases) | 0.30 (72 cases) |
| | | 0.25 (16 cases) | 0.15 (3 cases) | 0.12 (5 cases) | 0.25 (236 cases) |
| | | | | | 0.15 (18 cases) |
| | Manufacturer | Imaging Sciences International | Planmeca | Vatech; Sirona | Instrumentarium Dental; LargeV; Bondent Imaging; Carestream Health; Trophy; FUSSEN; PointNix; HDXWILL; GENORAY; SOREDEX; RAY; MEYER; |
| | Manufacturer's model | 17-19 | ProMax | PHT-6500; XG3D | OP300; HighRes3D; Bondream 3D-1020; CS 9300; K9500; Point Combi 500; DENTRI-C; PAPAYA 3D; Cranex3D; RAYSCAN; SS-X10010DPlus; SS-X9010DPro; SS-X9010DPro-3DE |
| | Tube voltage (kvp) | 80-120 | 80-90 | 70-100 | 70-120 |
| | Tube current (mA) | 5-12 | 4-14 | 3-10 | 4-13 |
| Dental abnormalities | Missing teeth (cases) | 613 | 497 | 181 | 137 |
| | Misalignment (cases) | 1141 | 1286 | 1009 | 314 |
| | Metal artifacts (cases) | 337 | 225 | 72 | 96 |

In demographic variables, the age is presented as average (range). In imaging protocols, the resolution is presented as a specific number (number of cases).

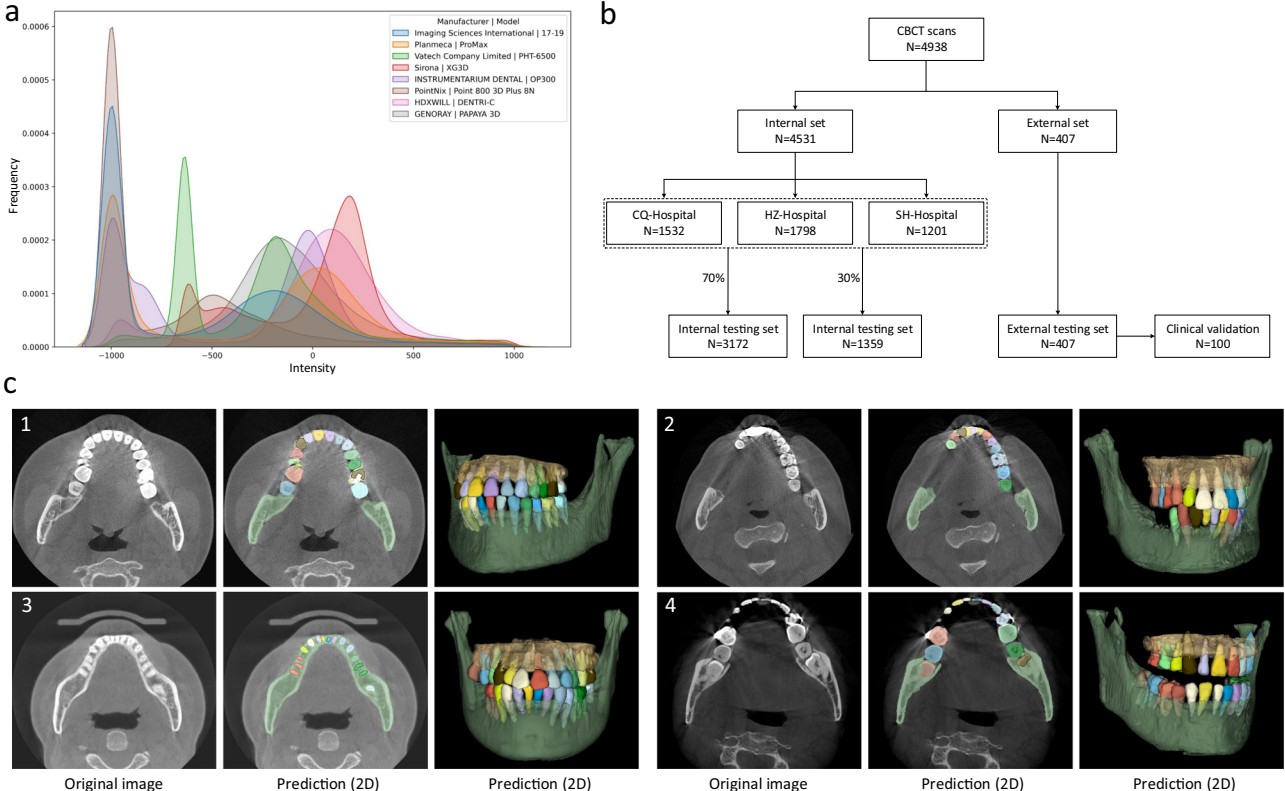

**Fig. 1 Data information and segmentation results in the multi-center CBCT dataset. a** The overall intensity histogram distributions of the CBCT data collected from different manufacturers. **b** The CBCT dataset consists of internal set and external set. The internal set collected from three hospitals is randomly divided into the training dataset and internal testing dataset. All 407 external CBCT scans, collected from 12 dental clinics, are used as external testing dataset, among which 100 CBCT scans are randomly selected for clinical validation by comparing the performance with expert radiologists. **c** Qualitative comparison of tooth and bone segmentation on the four center sets. The original CBCT images are shown in the 1st column, and the segmentation results in 2D and 3D views are shown in the 2nd and 3rd columns, respectively.

more than one kind of abnormality. Overall, this proof-of-concept study can fully mimic the heterogeneous environments in real-world clinical practice.

**Segmentation performance**. An overview of our AI system for tooth and alveolar bone segmentation is illustrated in Fig. 2. Given an input CBCT volume, the framework applies two concurrent branches for tooth and alveolar bone segmentation, respectively (see details provided in the "Methods" section). The segmentation accuracy is comprehensively evaluated in terms of three commonly used metrics, including Dice score, sensitivity, and average surface distance (ASD) error. Specifically, Dice is used to measure the spatial overlap between the segmentation result $R$ and the ground-truth result $G$, defined as Dice $= \frac{2|R \cap G|}{|R|+|G|}$. The sensitivity represents the ratio of the true positives to true positives plus false negatives. The distance metric ASD refers to the ASD of segmentation result $R$ and ground-truth result $G$.

Table 2 lists segmentation accuracy (in terms of Dice, sensitivity, and ASD) for each tooth and alveolar bone calculated on both the internal testing set (1359 CBCT scans from 3 known/seen centers) and external testing set (407 CBCT scans from 12 unseen centers). It can be observed that, on the internal testing set, our AI system achieves the average Dice score of 94.1%, the average sensitivity of 93.9%, and the average ASD error of 0.17 mm in segmenting individual teeth. The accuracy across different teeth is consistently high, although the performance on the 3rd molars (i.e., the wisdom teeth) is slightly lower than other teeth. This is reasonable, as many patients do not have the 3rd

molars. Also, the 3rd molars usually have significant shape variations, especially on the root area. The accuracy of our AI system for segmenting alveolar bones is also promising, with the average Dice score of 94.5% and the ASD error of 0.33 mm on the internal testing set.

Results on the external testing set can provide additional information to validate the generalization ability of our AI system on unseen centers or different cohorts. Specifically, from Table 2 we find that our AI system achieves an average Dice of 92.54% (tooth) and 93.8% (bone), sensitivity of 92.1% (tooth) and 93.5% (bone), and ASD error of 0.21 mm (tooth) and 0.40 mm (bone) on the external dataset. It indicates that the performance on the external set is only slightly lower than those on the internal testing set, suggesting high robustness and generalization capacity of our AI system in handling heterogeneous distributions of patient data. This is extremely important for an application developing for different institutions and clinical centers in real-world clinical practice.

As a qualitative evaluation, we show the representative segmentation produced by our AI system on both internal and external testing sets in Fig. 1c, where the individual teeth and surrounding bones are marked with different colors. We find that, although the image styles and data distributions vary highly across different centers and manufacturers, our AI system can still robustly segment individual teeth and bones to reconstruct 3D model accurately.

In clinical practice, patients seeking dental treatments usually suffer from various dental problems, e.g., missing teeth, misalignment, and metal implants. Accurate and robust segmentation of

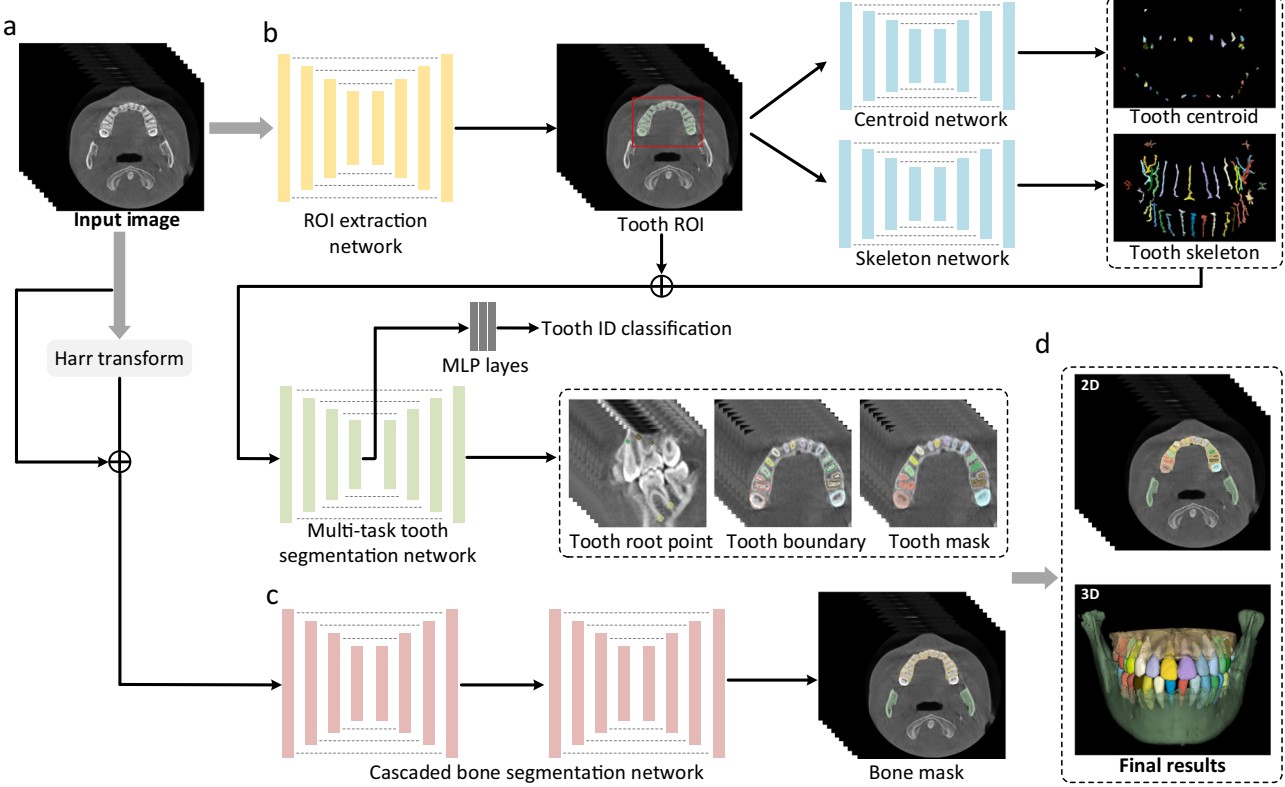

**Fig. 2 Overview of our proposed AI system for segmenting individual teeth and alveolar bones from CBCT images. a** The input of the system is a 3D CBCT scan. **b** The morphology-guided network is designed to segment individual teeth. **c** The cascaded network is used to extract alveolar bones. **d** The outputs of the model include the masks of individual teeth and alveolar bones.

**Table 2 Results of the individual teeth and bone segmentation on internal and external testing sets.**

| Tooth class | Internal testing set | | | External testing set | | |
|---|---|---|---|---|---|---|
| | Dice (%) | Sen (%) | ASD (mm) | Dice (%) | Sen (%) | ASD (mm) |
| Central incisor | 93.9 (79.4–96.2) | 94.7 (83.8–96.3) | 0.16 (0.09–0.27) | 92.6 (63.4–96.9) | 92.9 (65.8–97.5) | 0.23 (0.12–0.42) |
| Lateral incisor | 93.7 (68.5–96.6) | 92.8 (71.9–96.9) | 0.17 (0.07–0.35) | 92.4 (64.9–97.1) | 90.9 (60.2–95.4) | 0.21 (0.09–0.39) |
| Cuspid | 95.2 (82.9–97.6) | 93.9 (80.3–99.0) | 0.14 (0.05–0.21) | 94.2 (76.4–97.8) | 93.7 (75.9–98.6) | 0.17 (0.07–0.28) |
| 1st premolar | 95.0 (76.9–97.2) | 93.0 (75.3–96.8) | 0.15 (0.07–0.32) | 93.3 (61.4–96.9) | 91.7 (59.8–96.8) | 0.18 (0.10–0.35) |
| 2nd premolar | 94.9 (72.8–98.0) | 94.7 (76.9–97.2) | 0.16 (0.07–0.34) | 92.9 (70.5–96.7) | 90.5 (72.4–95.9) | 0.19 (0.08–0.48) |
| 1st molar | 94.6 (62.6–97.6) | 93.2 (60.8–97.5) | 0.18 (0.09–0.41) | 92.6 (68.8–97.4) | 91.9 (70.6–97.4) | 0.24 (0.10–0.41) |
| 2nd molar | 93.4 (67.2–98.2) | 90.7 (66.8–94.7) | 0.19 (0.08–0.38) | 91.7 (63.9–97.0) | 91.7 (66.7–96.0) | 0.23 (0.07–0.56) |
| 3nd molar | 91.5 (52.9–95.8) | 92.7 (58.9–96.7) | 0.21 (0.13–0.72) | 91.3 (53.7–96.4) | 90.6 (51.0–96.2) | 0.28 (0.14–0.94) |
| Average | 94.1 | 93.9 | 0.17 | 92.5 | 92.1 | 0.21 |
| Maxillary bone | 94.1 (76.9–96.9) | 93.5 (74.1–95.8) | 0.35 (0.18–0.84) | 93.0 (57.9–95.4) | 92.8 (49.3–95.4) | 0.47 (0.18–0.96) |
| Mandible bone | 94.8 (80.3–97.3) | 94.2 (83.0–97.4) | 0.29 (0.13–0.77) | 94.5 (67.7–97.8) | 93.9 (72.5–96.9) | 0.33 (0.12–0.76) |
| Average | 94.5 | 93.8 | 0.33 | 93.8 | 93.5 | 0.40 |

CBCT images for these patients is essential in the workflow of digital dentistry. Figure 3 presents the comparison between segmentation results (in terms of Dice score and sensitivity) produced by our AI system on healthy subjects and also the patients with three different dental problems. By regarding those results on the healthy subjects as the baseline, we can observe that our AI system can still achieve comparable performance for the patients with missing and misaligned teeth, while slightly reduced performance for the patients with metal implants (i.e., for the CBCT images with metal artifacts). Also, in Fig. 4, we visualize both tooth and bone segmentation results on representative CBCT images with dental abnormalities (Fig. 4a–f) and normal CBCT images (Fig. 4g, h). Although metal artifacts introduced by dental fillings, implants, or metal

crowns greatly change the image intensity distribution (Fig. 4a, b), our AI system can still robustly segment individual teeth and bones even with very blurry boundaries. In addition, by observing example segmentation results for the CBCT images with missing teeth (Fig. 4c, d) and/or misalignment problems as shown in Fig. 4e, f, we can see that our AI system still achieves promising results, even for the extreme case with an impacted tooth as highlighted by the red box in Fig. 4e.

**Ablation study**. To validate the effectiveness of each important component in our AI system, including the skeleton representation and multi-task learning scheme for tooth segmentation, and the harr filter transform for bone segmentation, we have

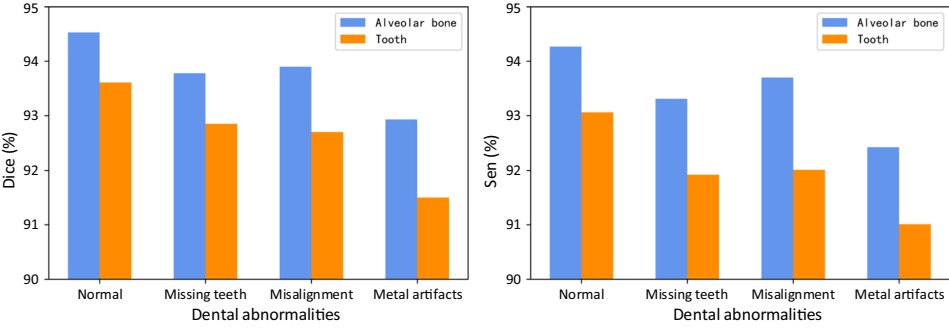

**Fig. 3** Segmentation performance of the CBCT scans with different dental abnormalities, including the Dice and the sensitivity.

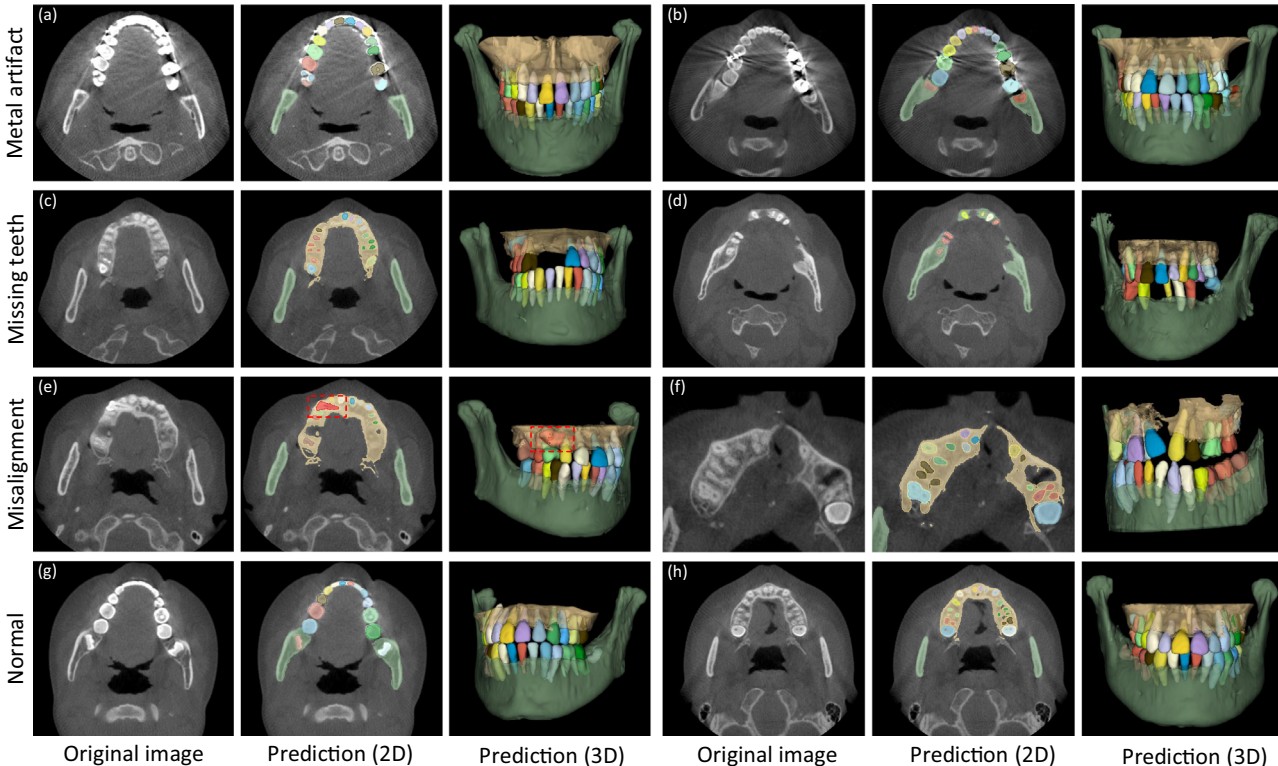

**Fig. 4 Typical tooth and bone segmentation results from the CBCT images with dental abnormalities.** Images with metal artifacts (**a**, **b**), missing teeth (**c**, **d**) and misalignment problems (**e**, **f**), and without dental abnormality (**g**, **h**).

conducted a set of ablation studies shown in Supplementary Table 2 in the Supplementary Materials. First, for the tooth segmentation task, we train three competing models, i.e., (1) our AI system (AI), (2) our AI system without skeleton information (AI (w/o S)), and (3) our AI system without the multi-task learning scheme (AI (w/o M)). It can be seen that AI (w/o S) and AI (w/o M) show relatively lower performance in terms of all metrics (e.g., Dice score of 2.3 and 1.4% on the internal set, and 1.4 and 1.1% on external set), demonstrating the effectiveness of the hierarchical morphological representation for accurate tooth segmentation. Moreover, the multi-task learning scheme with boundary prediction can greatly reduce the ASD error, especially on the CBCT images with blurry boundaries (e.g., with metal artifacts). Next, for the alveolar bone segmentation task, we compare our AI system with the model without harr filter enhancement (AI (w/o H)). Our AI system can increase Dice score by 2.7% on internal testing set, and 2.6% on external testing set, respectively. The improvements are significant, indicating enhancing intensity contrast between alveolar bones and soft tissues to allow the bone segmentation network to learn more accurate boundaries.

**Comparison with other methods**. To show the advantage of our AI system, we conduct three experiments to directly compare our AI system with several most representative deep-learning-based tooth segmentation methods, including ToothNet[24], MWTNet[27], and CGDNet[28]. Note that, ToothNet is the first deep-learning-based method for tooth annotation in an instance-segmentation fashion, which first localizes each tooth by a 3D bounding box, followed by the fine-grained delineation. MWTNet is a semantic-based method for tooth instance segmentation by identifying boundaries between different teeth. CGDNet detects each tooth's center point to guide their delineation, which reports the state-of-the-art segmentation accuracy. Notably, all these three competing methods are designed solely for tooth segmentation, as there is no study in the literature so far for jointly automatic alveolar bone and tooth instance segmentation.

| Table 3 Quantitative comparison between our AI system and two expert radiologists (tested on 100 CBCT scans randomly selected from external set). | | | |
|---|---|---|---|
| **Model** | **Expert-1** | **Expert-2** | **AI system** |
| Dice (tooth) (%) | 91.9 | 92.1 | 92.4 |
| Dice (bone) (%) | 92.7 | 93 | 93.3 |
| Time (min) | 147 | 160 | 0.23 |
| Time (min) (with AI assistance) | 4.3 | 4.9 | – |
| Modification scans | 12/100 | 12/100 | – |

Considering that these competing methods are trained and evaluated with very limited data in their original papers, we conduct three new experiments under three different scenarios for comprehensive comparison with our method. Specifically, we train these competing models, respectively, by using (1) a small-sized training set (100 CBCT scans), (2) a small-sized training set with data argumentation techniques (100+ CBCT scans), and (3) a large-scale training set with 3172 CBCT scans. Corresponding segmentation results on the external dataset are provided in Supplementary Table 3 in the Supplementary Materials. From Supplementary Table 3, we can have two important observations. First, our AI system consistently outperforms these competing methods in all three experiments, especially for the case when using small training set (i.e., 100 scans). These results show the advance of various strategies we proposed. For example, instead of simply localizing each tooth by points or bounding boxes as used in these competing methods, our AI system learns a hierarchical morphological representation (e.g., tooth skeleton, tooth boundary, and root apices) for individual teeth often with varying shapes, and thus can more effectively characterize each tooth even with blurring boundaries using small training dataset. Second, for all methods (including our AI system), the data argumentation techniques (100+) can consistently improve the segmentation accuracy. However, compared with the large-scale real-clinical data (3172 CBCT scans), the improvement is not significant. This further demonstrates the importance of collecting large-scale dataset in clinical practice.

In summary, compared to the previous deep-learning-based tooth segmentation methods, our AI system has three aspects of advantage. First, our AI system is fully automatic, while most existing methods need human intervention (e.g., having to manually delineate foreground dental ROI) before tooth segmentation. Second, our AI system has the best tooth segmentation accuracy because of our proposed hierarchical morphological representation. Third, to the best of our knowledge, our AI system is the first deep-learning work for joint tooth and alveolar bone segmentation from CBCT images.

**Comparison with expert radiologists.** To verify the clinical applicability of our AI system for fully automatic tooth and alveolar bone segmentation, we compare its performance with expert radiologists on 100 CBCT scans randomly selected from the external set. We enroll two expert radiologists with more than 5 years of professional experience. Note that these two expert radiologists are not the people for ground-truth label annotation.

The comparison results are summarized in Table 3. It can be seen that, in terms of segmentation accuracy (e.g., Dice score), our AI system performs slightly better than both expert radiologists, with the average Dice improvements of 0.55% (expert-1) and 0.28% (expert-2) for delineating teeth, and 0.62% (expert-1) and 0.30% (expert-2) for delineating alveolar bones. Accordingly, we also compute corresponding $p$ values to validate whether the improvements are statistically significant. The

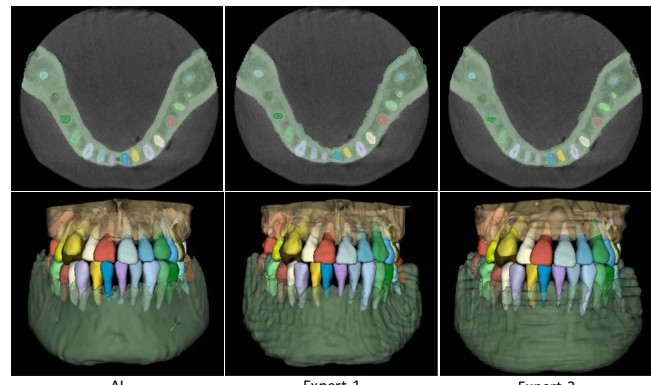

**Fig. 5** Qualitative segmentation results produced by our AI system and two expert radiologists.

statistical significance is defined as 0.05. Specifically, for tooth segmentation, the paired $p$ values are 2e−5 (expert-1) and 7e−3 (expert-2). And for alveolar bone segmentation, the paired $p$ values are 1e−3 (expert-1) and 9e−3 (expert-2). All $p$ values are smaller than 0.05, indicating that the improvements over manual annotation are statistically significant. Another observation is worth mentioning that the expert radiologists obtained a lower accuracy in delineating teeth than alveolar bones (i.e., 0.79% by expert-1 and 0.84% by expert-2 in terms of Dice score). This is because teeth are relatively small objects, and neighboring teeth usually have blurry boundaries, especially at the interface between upper and lower teeth under a normal bite condition. Also, due to the above challenge, the segmentation efficiency of expert radiologists is significantly worse than our AI system. Table 3 shows that the two expert radiologists take 147 and 169 min (on average) to annotate one CBCT scan, respectively. In contrast, our AI system can complete the entire delineation process of one subject within only a couple seconds (i.e., 17 s). Besides quantitative evaluations, we also show qualitative comparisons in Fig. 5 to check visual agreement between segmentation results produced by our AI system and expert radiologists. It can be seen that the 3D dental models reconstructed by our AI system have much smoother surfaces compared to those annotated manually by expert radiologists. These results further highlight the advantage of conducting segmentation in the 3D space (i.e., by our AI system) rather than 2D slice-by-slice operations (i.e., by expert radiologists).

**Clinical improvements.** Besides direct comparisons with experts from both aspects of segmentation accuracy and efficiency, we also validate the clinical utility of our AI system, i.e., whether this AI system can assist dentists and facilitate clinical workflows of digital dentistry. To this end, we roughly calculate the segmentation time spent by the two expert radiologists under assistance from our AI system. Specifically, instead of fully manual segmentation, the expert radiologists first apply our trained AI system to produce initial segmentation. Then, they check the initial results slice-by-slice and perform manual corrections when necessary, i.e., when the outputs from our AI system are problematic according to their clinical experience. Therefore, the overall work time includes the time verifying and updating segmentation results from our AI system.

The corresponding results are summarized in Table 3. Without assistance from our AI system, the two expert radiologists spend about 150 min on average to manually delineate one subject. In contrast, with the assistance of our AI system, the annotation time is dramatically reduced to less than 5mins on average, which is ~96.7% reduction in segmentation time. A paired $t$-test shows

statistically significant improvements with $P_1 = 3.4 \times 10^{-13}$ and $P_2 = 5.4 \times 10^{-15}$, with respect to the two expert radiologists, respectively. Also, it is worth noting that the expert radiologists accepted most of the fully automatic prediction results produced by our AI system without any modification, except only 12 out of the 100 CBCT scans requiring extra-human intervention. For example, the predicted tooth roots may have a little over- or under-segmentation. Additional refinements can make the dental diagnosis or treatments more reliable. Our clinical partners have confirmed that such performance is fully acceptable for many clinical and industrial applications, e.g., doctor-patient communications and treatment planning for orthodontics or dental implants, indicating the high clinical utility of our AI system.

## Discussion

The 3D information of teeth and surrounding alveolar bones is essential and indispensable in digital dentistry, especially for orthodontic diagnosis and treatment planning. In this study, we develop the first clinically applicable deep-learning-based AI system for fully automatic tooth and alveolar bone segmentation. We have validated our system in real-world clinical scenarios with very large internal (i.e., 1359 CBCT scans) and external (i.e., 407 CBCT scans) datasets, and obtained high accuracy and applicability as confirmed by various experiments.

One of the key attributes of our AI system is full automation with good robustness. Most conventional methods[5–7] are semi-automatic, i.e., they typically requiring additional user annotation to first identify individual teeth before delineating the tooth boundary (e.g., using level set or graph cut). For example, Gan et al.[7] have developed a hybrid level set based method to segment both tooth and alveolar bone slice-by-slice semi-automatically. Note that a starting slice and seed point of each tooth should be manually selected for the detection of individual tooth regions, which is time-consuming and laborious in clinical practice. Recently, many deep learning-based methods[24–30] with various network architectures have been designed. Given a predefined ROI, most of these learning-based methods can segment teeth automatically. However, ROIs often have to be located manually in the existing methods (e.g., ToothNet[24] and CGDNet[28]), thus, the whole process for teeth segmentation from original CBCT images is not fully automatic. Instead, our AI system is fully automatic, and the whole pipeline can be run without any manual intervention, including the dental ROI localization, tooth segmentation, and alveolar bone segmentation with input of original CBCT images. To improve model robustness and generalizability, some existing methods also have attempted to address the challenging cases with metal artifacts. For example, a dense ASPP module has been designed in CGDNet[28] for this purpose, and achieved leading performance, but it only tested on a very small dataset with 8 CBCT scans. Our AI system can more robustly handle the challenging cases than CGDNet, as demonstrated by the comparisons in Supplementary Table 3, using either small-size dataset or large-scale dataset. This is mainly due to the two proposed complementary strategies for explicitly enhancing the network learning of tooth geometric shapes in the CBCT images (especially with metal artifacts or blurry boundaries). First, we explicitly capture tooth skeleton information to provide rich geometric guidance for the downstream individual tooth segmentation. Second, we use tooth boundary and root landmark prediction as an auxiliary task for tooth segmentation, thus explicitly enhancing the network learning at tooth boundaries even with limited intensity contrast (e.g., metal artifacts). It is worth noting that the relationship between teeth and alveolar bones is critical in clinical practice, especially in orthodontic treatment, because the tooth root apices cannot penetrate the

surrounding bones during tooth movement. Moreover, we also introduce a filter-enhanced (i.e., Harr transform) cascaded network for accurate bone segmentation by enhancing intensity contrasts between alveolar bones and soft tissues. Such combinations of data-driven and knowledge-driven approaches have demonstrated promising performance in particular tasks, such as image decomposition[33], tissue segmentation[34], and depth estimation[35]. Hence, our system is fully automatic with good robustness, which takes as input the original 3D CBCT image and automatically produces both the tooth and alveolar bone segmentations without any user intervention.

Another important contribution of this study is that we have conducted a series of experiments and clinical applicability tests on a large-scale dataset collected from multi-center clinics, demonstrating that deep learning has great potential in digital dental dentistry. Previous studies have mostly focused on algorithm modifications and tested on a limited number of single-center data, without faithful verification of model robustness and generalization capacity. For example, Cui et al.[24] have applied an instance segmentation method (Mask R-CNN[36]) from the computer vision community to tooth instance segmentation and achieved an average Dice score of 93.3% on 8 testing CBCT scans. However, the performance on the multi-center external dataset has not been validated, i.e., not tested on the diverse and unseen data scanned with different image protocols, scanner brands, or parameters. Recently, the data argumentation techniques have been widely used to improve model robustness in medical image analysis[37]. As shown in Table 3, by applying the data argumentation techniques (e.g., image flip, rotation, random deformation, and conditional generative model[38]), the segmentation accuracy of different competing methods indeed can be boosted. But the improvements are limited compared with the large-scale dataset collected from real-world clinics. It is mainly because such a small-sized set of real data, as well as the synthesized data (using data argumentation methods), cannot completely cover the dramatically varying image styles and dentition shape distributions in clinical practice. And the large-scale, multi-center, and real-clinical data collected in this study can effectively address this issue. Specifically, as shown in Fig. 1a, the acquired images present large style variations across different centers in terms of imaging protocols, scanner brands, and/or parameters. In addition, as reported by the oral health survey[39,40], the dentition distributions (i.e., tooth size) can be a little different across people from different regions. More importantly, since all the CBCT images are scanned from patients with dental problems, different centers may have large different distributions in dental abnormalities, which further increases variations in tooth/bone structures (i.e., shape or size). The results presented in Supplementary Table 3 strongly support the observation that a large-scale and heterogeneous dataset is essential for building a robust and generalizable deep learning system in clinics. The experimental observations in Fig. 3 and Table 2 have also shown that our AI system can produce consistent and accurate segmentation on both internal and external datasets with various challenging cases collected from multiple unseen dental clinics. Furthermore, extensive clinical validations and comparisons with expert radiologists have verified the clinical applicability of our AI system, especially in greatly reducing human efforts in manual annotation and inspection of the 3D tooth and alveolar bone segmentations.

In addition, to validate the automation, robustness, and clinical applicability of our AI system, we also explore the clinical knowledge embedded in the large-scale CBCT dataset, i.e., the trajectory of tooth volume and density changes with ages of participants. It is worth noting that the trajectory curves are computed from the ground truth annotation, instead of our AI system prediction, which is more convincing from clinical

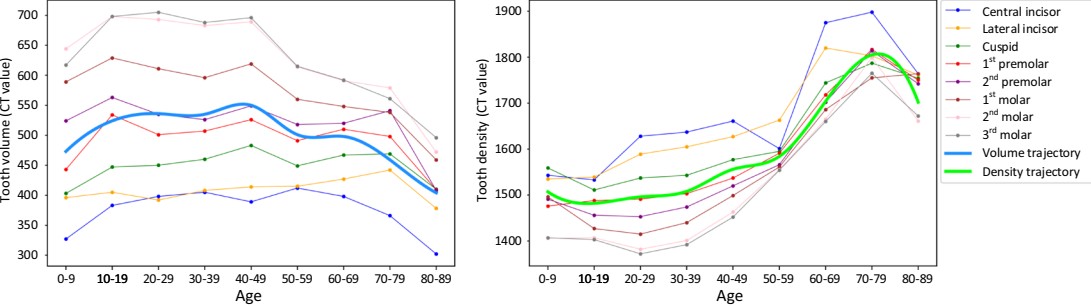

**Fig. 6** The changing curves of tooth volumes and intensities over different ages of patients.

perspectives. The corresponding results are shown in Fig. 6, where the volume and density of each tooth are quantified at different age ranges from all collected CBCT scans (i.e., internal and external datasets). From Fig. 6, we can have a consistent observation that the volumes of all teeth increase significantly from Children (0–9 years old) to Youth (10–19 years old), as this period is the "mixed dentition" time that children usually lose their deciduous teeth (smaller in size) and gain permanent adult teeth. Moreover, the volume of tooth rapidly decreases after 50 years old due to tooth wear or broken, especially for molar teeth. On the other hand, the trajectories of densities for different teeth also have consistent patterns, i.e., gradual increase during the period of 30–80 years old while obvious decrease at 80–89 years old. These imaging findings are consistent with the existing clinical knowledge, which has shown that the tooth enamel changes over time, and it may disappear after 80 years old due to day-to-day wear and tear of teeth. With the volume and density changing curves as shown in Fig. 6, an interesting phenomenon can be observed that there is a peak in the volume trajectory curve for middle-aged patients. The potential reasons are two-fold. First, as reported, there is a significant tooth size discrepancy across people from different regions[39,40]. And in this study, our dataset (i.e., internal and external sets) is mainly collected from three places (i.e., Chongqing, Hangzhou, and Shanghai), where their tooth size distributions may be slightly different and thus lead to the peak in the volume trajectory curve for middle-aged patients. The second reason may be that all the CBCT images are collected from patients seeking different dental treatments in hospitals, which may also produce peak value in the volume trajectory curve. Generally, such studies on tooth development trajectories could facilitate a better understanding of dental diseases and healthcare. In the future, we plan to collect larger data from more centers, and calculate the tooth volume and intensity trajectories with different scenarios, including inter- and intra-different regions, and before and after dental treatments.

Although this work has achieved overall promising segmentation results, it still has flaws in reconstructing the detailed surfaces of the tooth crown due to the limited resolution of CBCT images (i.e., 0.2–0.6 mm). In clinics, the 3D dental model scanned by the intra-oral scanner is often acquired to represent the tooth crown surface with much higher resolution (0.01–0.02 mm), which is helpful in tooth occlusion analysis but without tooth root information. Thus, it is valuable to leverage the intra-oral scans to improve the tooth crown shapes reconstructed from CBCT images. In future work, as a post-processing step of our current method, we will collect some paired intra-oral scans and combine it with the CBCT segmentation results to build a complete 3D tooth and alveolar bone model with a high-resolution tooth crown shape. This will lead to a more accurate AI system for digital dentistry.

In conclusion, this study proposes a fully automatic, accurate, robust, and most importantly, clinically applicable AI system for 3D tooth and alveolar bone segmentation from CBCT images, which has been extensively validated on the large-scale multi-center dataset of dental CBCT images. It also suggests that combing artificial intelligence and dental medicine would lead to promising changes in future digital dentistry.

## Method

**Image pre-processing**. This study was approved by the Research Ethics Committee in Shanghai Ninth People's Hospital and Stomatological Hospital of Chongqing Medical University. Due to the retrospective nature of this study, the informed consent was waived by the relevant IRB. Before feeding a 3D CBCT image into the deep learning network, we pre-process it with the following steps. First, as the physical resolution of our collected CBCT images varies from 0.2 to 1.0 mm, all CBCT images are normalized to an isotropic resolution of $0.4 \times 0.4 \times 0.4$ mm$^3$, considering the balance between computational efficiency and segmentation accuracy. For example, if the resolution is higher than 0.4 mm, down-sampling is introduced; otherwise, up-sampling is applied on the 3D CBCT images. Additionally, following the standard protocol of image processing in deep learning, the voxel-wise intensities are normalized to the interval [0, 1]. Moreover, to reduce the effect of extreme values, especially at the area of metal artifacts, we clip intensity values of each CBCT scan to [0, 2500] before intensity normalization.

**Model implementation**. Figure 2 presents the overview of our deep-learning-based AI system, including a hierarchical morphology-guided network to segment individual teeth and a filter-enhanced network to extract alveolar bony structures from the input CBCT images. We elaborate each of these two networks in this subsection, and the detailed network architectures are shown in Supplementary Materials (Supplementary Figs. 2–5).

Considering the field-of-view in 3D CBCT image usually captures the entire maxillofacial structures, the dental area is relatively small. In this sense, we first apply an encoder-decoder network to automatically segment the foreground tooth for dental area localization. Note that it is a binary segmentation task without separating different teeth. As shown in Fig. 2, we directly employ V-Net[41] in this stage to obtain the ROI. Specifically, due to the limitation of GPU memory, we randomly crop patches of size $256 \times 256 \times 256$ from the CBCT image as inputs. In the network training stage, the binary cross-entropy loss is utilized to supervise the probability map outputted by the last convolutional layer.

After obtaining the dental ROI, we use our previously-developed hierarchical morphology-guided network[30] to make automatic and accurate segmentation of individual teeth. This a two-stage network first detects each tooth and represents it by the predicted skeleton, which can stably distinguish each tooth and capture the complex geometric structures. Then, based on the output of the first step, a multi-task learning network for single tooth segmentation is introduced to predict each tooth's volumetric mask by simultaneously regressing the corresponding tooth apices and boundaries. The design of the method is natural, as it can properly represent and segment each tooth from background tissues, especially at the tooth root area where accurate segmentation is critical in orthodontics to ensure that the tooth root cannot penetrate the surrounding bone during tooth movements. The overview network architecture is shown in Fig. 2. Specifically, the centroid and skeleton detection networks in the first step are all V-Net[41] structures with two output branches. One is the 3D offset map (i.e., 3D vector) pointing to the corresponding tooth centroid points or skeleton lines, and the other branch outputs a binary tooth segmentation mask to filter out background voxels in the 3D offset maps. With the predicted tooth centroid points and skeletons, a fast clustering method[42] is first implemented to distinguish each tooth based on the spatial centroid position, and simultaneously recognize tooth numbers. Then, each detected tooth can be represented by its skeleton. In the second step of single tooth segmentation, the three-channel inputs to the multi-task tooth segmentation network are the patches cropped from the tooth centroid map, the skeleton map, and the tooth ROI images, respectively. The size of each channel is $96 \times 96 \times 96$. As shown in Fig. 2, a V-Net network architecture with multiple task-specific outputs is used to predict the mask of each individual tooth. Note that, in the multi-task tooth

segmentation network, the encoder part is followed by a max-pooling layer and three fully-connected layers to identify the category of each input tooth patch, based on the FDI World Dental Federation notation system[43]. In the training stage, we respectively adopt binary cross-entropy loss to supervise the tooth segmentation, and another L2 loss to supervise the 3D offset, tooth boundary, and apice prediction.

The alveolar bone segmentation framework is developed based on a boundary-enhanced neural network, which aims to directly extract midface and mandible bones from input 3D CBCT image. Specifically, as shown in Fig. 2, we first utilize harr transform[44] to process the CBCT image, where the intensity contrast around bone boundaries can be significantly enhanced. Then, with the filtered image, we combine it with the original CBCT image, and feed them into a cascaded V-Net[41]. The input of the original and filtered images are the cropped patches with a dimension of $256 \times 256 \times 256$ considering the limitation the GPU memory limitation. The output of the network is a 3-channel mask, with the same size as the input patch, indicating probabilities of each voxel belonging to the background, midface bone, and mandible bone, respectively. To train the network, we adopt the cross-entropy loss to supervise the alveolar bone segmentation. Note that, in the inference time, a post-processing step is employed to merge the predicted bone and tooth masks. For example, if a voxel is simultaneously predicted as bone and tooth, we will compare the probabilities predicted by the bone and tooth segmentation networks, and choose the label with a larger probability as the final prediction.

**Training details**. The framework was implemented in PyTorch library[45], using the Adam optimizer to minimize the loss functions and to optimize network parameters by back propagation. A learning rate of 0.001 and a mini batch size of 1 were used in the tooth and alveolar bone segmentation network. At the end of each training epoch, we computed the loss on the validation dataset to determine the network convergence. If the model performance on the validation dataset remained unchanged for 5 epochs, we considered that the training process was converged and could be stopped. All deep neural networks were trained with one Nvidia Tesla V100 GPU.

**Reporting summary**. Further information on research design is available in the Nature Research Reporting Summary linked to this article.

## Data availability
The authors declare that partial data (i.e., 50 raw data of CBCT scans collected from dental clinics) will be released to support the results in this study (link: https://pan.baidu.com/s/1LdyUA2QZvmU6ncXKl_bDTw, password:1234), with permission from respective data centers. The full datasets are protected because of privacy issues and regulation policies in hospitals. All requests about the software testing, comparison and evaluation can be sent to the first author (Z.C., Email: cuizm.neu.edu@gmail.com). All requests will be promptly reviewed within 15 working days.

## Code availability
The code of this system would be accessible (https://pan.baidu.com/s/194DfSPbgi2vTIVsRa6fbmA, password:1234). It should be used for academic research only.

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

## Acknowledgements

This work was supported in part by National Natural Science Foundation of China (grant number 62131015), Science and Technology Commission of Shanghai Municipality (STCSM) (grant number 21010502600), and The Key R&D Program of Guangdong Province, China (grant number 2021B0101420006).

## Author contributions

In this study, Z.C., Y.F., L.M., C.L. and D.S. designed the method, and drafted the manuscript. Z.C., Y.F., and L.M. wrote the code. B.Z., B.Y., Y.L., Y.Z., Z.D., and M.Z. collected and processed the dataset. J.L., Y.S., L.M., and J.H. provided statistical analysis and interpretation of the data. D.S. coordinated and supervised the whole work. All authors were involved in critical revisions of the manuscript, and have read and approved the final version.

## Competing interests

The authors declare no competing interests.
