## [Peer Review File · Nature Communications]

Reviewers' Comments:

Reviewer #1:

Remarks to the Author:

This paper deals with the important problem of digital dentistry, which automatically segment teeth and alveolar bone in dental CBCT projections. This paper used a huge data set (4938 CBCT scans from 4215 patients) to apply deep the proposed deep learning method. The effort to gather such huge data is astounding. The results appear good. However, it is difficult to say that the proposed method is better than the existing deep learning-based tooth segmentation. The authors' comments about previous works are misleading. The existing methods are also fully automatic and can deal with metal artifacts and missing teeth. Also, there is no reason to believe that the proposed method is effective for handling metal artifacts. This paper lacks a detailed analysis of why the proposed method is good. Dental CBCT image quality depends on factors (e.g., tube voltages and tube current) that influence patient radiation dose. This paper mentions only hospitals that obtain CBCT data, and does not provide information on the types of CBCT machines used and radiation dose (tube current, tube voltage, etc.). It seems that the only contribution of this paper is that the amount of data is very large.

Reviewer #2:

Remarks to the Author:

This paper proposes an automated tooth segmentation method in CBCT images. It first extracts the ROI. Next, it detects skeleton and then predicts tooth apices and boundary. The tooth region is segmented by faster clustering using the centroid. First of all, I would like to congratulate you on your excellent performance.

There are a few points before the publication.

- 1) Does it recognize tooth numbers? Maybe no. I am a little confused because the experimental results show the comparison.
- 2) Show the data distribution of the abnormalities in the training and test dataset.
- 3) Discuss the reason why this method achieves better results for abnormalities.

Reviewer #3:

Remarks to the Author:

This paper presented a deep learning-based system for automatic tooth and alveolar bone segmentation from CBCTs. The authors have collected a large CBCT dataset from 4215 patients (with 4938 CBCT scans) of 15 different centers. The proposed model reports a performance gain against experienced radiologists in terms of the average Dice similarity coefficient. The experimental results are plausible. The major concerns are as follows:

1. The proposed model was trained and validated on a large dataset. How about the performance gains compared with the existing deep learning-based tooth segmentation model learned from limited data? The data augmentation techniques have been widely used in medical image segmentation, where the generative model is effective in generating a large dataset with annotations [a]. It would be helpful to discuss the advantages of large data.
[a] Zhao, Amy, Guha Balakrishnan, Frédo Durand, John V. Guttag and Adrian V. Dalca. "Data Augmentation Using Learned Transformations for One-Shot Medical Image Segmentation." 2019 IEEE/CVF Conference on Computer Vision and Pattern Recognition (CVPR) (2019): 8535-8545.
2. What is the image styles and data distributions variations across different centers and manufacturers? It is not clear on the capacity of the proposed system on CBCTs of different styles and distributions.
3. The ground truth annotations were obtained by the manual annotation of senior radiologists with ten or more years of professional experience. The manual annotation is time-consuming and

prone to inter-practitioner's variations. How about the time complexity of the ground truth annotation and the scheme to avoid inter-practitioner's variations regarding the hierarchical data annotation method?

4. As shown in Fig. 7, the output mask of the Cascaded bone segmentation network includes the tooth. It is unclear whether the mask of alveolar bones was obtained directly from the end-to-end model or by some postprocessing, e.g., Boolean operation with the tooth segmentation.

5. The proposed system was used to validate the automation and clinical applicability. Fig. 6. illustrates the curves of tooth volumes and intensities over different patient ages regarding different types of teeth. How about the difference between the curves computed from the automatic segmentation and manual annotation?

6. The Harr transform was used for image enhancements for bone segmentation. Why is such transform not used for tooth segmentation?

Since the learned convolutional kernels were used as image filters in feature extraction, it is unclear whether the Harr transform-like preprocessing is necessary for the image segmentation framework.

7. What is the input of the multi-task tooth segmentation network? As shown in Fig. 7, the tooth skeleton and centroid served as the input of the segmentation network. Whether the ROI features was used in tooth segmentation?

8. The proposed system performed the segmentation of teeth and alveolar bones better than expert radiologists. Does the slight performance gain have statistical significance? How about an annotation difference between the system and experts regarding the slight improvements?

Reviewer #4:

Remarks to the Author:

Nature Communications – Review of MS NCOMMS-21-40797

Title: A fully automatic, clinically applicable AI system for tooth and alveolar bone segmentation from cone-beam CT images

Review – General Comments:

Please note that my comments are from a clinician's perspective. In light of that, this manuscript describes an exciting clinical complement. The group earns applause for creating a useful, validated tool for clinicians as well as for engaging a multidisciplinary team in the system's development.

The inclusion of Figure 6 showing volume and density changes by tooth across different ages is interesting and might be the basis for another publication, perhaps in the dental literature. It would be interesting to have the authors conjecture why there is a peak in the volume trajectory curve for middle-aged patients.

It might be useful in future publications describing the system to address the diversity of the population that formed the basis for analysis. For instance, are all of the patients from a single culture, is a breadth of minorities included in the data set, etc.

It might be useful to somewhere add the definition of "DICE" for those clinicians who are not familiar with the term and the significance of the values that your report.

Throughout the document there is inconsistent use of () around e.g., and i.e.

The last lines of the introduction section address numbers of images used to validate the AI system. The numbers there differ from those used in the abstract and may be confusing, without further clarification, to the readers. I suggest that you reword those sentences to capitalize (and remind the readers) of the 4,215 patient records and nearly 5,000 CBCT scans you had access to and used.

More Specific comments:

- Line 33: perhaps the word 'speedup' could be replaced by 'faster'
- Line 40: generally, in clinical use, the term dental implant is used instead of dental implanting.
- Line 43: generally, in clinical use, the line might read "...X-rays, 3D Intra-Oral scans, and ...(CBCT) images."
- Line 62: replace 'changing appearances' to 'variations in structures'
- Line 64: change 'small-sized data' to 'small-sized data sets'
- Line 83: change 'implanting' to 'implants' and 'we collected 4,215' to 'we collected data of 4,215'

- Line 93: change 'CBCT' to 'of the CBCT'
- Line 107: 'Method' should be 'Methods'
- Line 124: Perhaps 'segment' would be a better word than 'extract'
- Line 142: Here you state that the radiologists had more than 5 years' experience. Elsewhere in the document you state that they had 10 years' experience. Are they different radiologists? Perhaps some clarification would be useful to avoid confusion.
- Line 171: You point out that 12 of 100 CBCT scans required extra-human intervention. It would be useful to know why that intervention was necessary.
- Line 178: 'Discussions' should be 'Discussion'
- Line 219: Perhaps better to replace 'become firmer' with 'changes'.

Summary: All in all, this was a great paper to read. The clinical relevance is good and offers great potential to improve clinical decisions as well as to improve the ability to assess treatment outcomes qualitatively and quantitatively over time. I look forward to seeing a similar report in the dental literature!

Reviewer: E. Dianne Rekow, DDS, PhD

Point-to-point Responses to Review Comments

Manuscript ID: NCOMMS-21-40797

Paper Title: A fully automatic, clinically applicable AI system for tooth and alveolar bone segmentation from cone-beam CT images

Dear Editor and Reviewers,

Thank you for your insightful comments. We are encouraged for receiving your positive feedbacks consistently, especially those affirmations regarding clinical meaningfulness (R1, R4), excellent performance of this work (R1, R2, R3, R4), and potential usability in improving clinical decisions (R4). These detailed comments and constructive suggestions indeed have greatly helped us improve the presentation and experimental analyses of our method more rigorously and comprehensively.

We have revised our paper according to review comments and prepared a revised version with changes highlighted in blue. The point-to-point response to each comment, along with respective revision in our paper, are summarized below:

To Reviewer #1:

This paper deals with the important problem of digital dentistry, which automatically segment teeth and alveolar bone in dental CBCT projections. This paper used a huge data set (4938 CBCT scans from 4215 patients) to apply the proposed deep learning method. The effort to gather such huge data is astounding. The results appear good.

Summary of Responses below: We'd like to thank the reviewer for encouraging comments, especially the clinical meaning of our research. According to review comments, we have thoroughly revised revised our manuscript by adding more experiments and correcting those misleading descriptions. The response to each specific comment is provided below.

Comment #1: However, it is difficult to say that the proposed method is better than the existing deep learning-based tooth segmentation.

Response: To address this comment, we have included three new experiments to directly compare our AI system with several most representative deep-learning-based tooth segmentation methods from the literature (i.e., ToothNet [1], MWTNet [2], and CGDNet [3]) by using the large-scale dataset as employed in our paper. Note that, ToothNet is the first deep-learning-based method for tooth annotation in an instance-segmentation fashion, which first localizes each tooth by a 3D bounding box, followed by the fine-grained delineation. MWTNet performs semantic-based tooth segmentation by identifying boundaries between different teeth. CGDNet detects each tooth's center point to guide their delineation, which reports the state-of-the-art segmentation accuracy. Notably, all these three competing methods are designed solely for tooth segmentation, as there is no study in the literature so far for jointly automatic alveolar bone and tooth instance segmentation.

Considering that these competing methods were trained and evaluated with very limited data in their original papers, we conduct three new experiments under three different scenarios for comprehensive and convincing comparisons between these competing methods and our method. Specifically, we train these competing models, respectively, by using 1) a small-sized training set (100 CBCT scans), 2) a small-sized training set with data argumentation techniques (100+ CBCT scans), and 3) a large-scale training set with 3172 CBCT scans. Their corresponding segmentation results are provided in **Table R1** below for the convenience of review. From Table R1, we can have two important observations:

1. For all methods (including our AI system), the tooth segmentation accuracy is consistently increased by a relatively large margin after extending the size of the training set. This further confirms the importance of collecting large-scale diverse dataset for tooth segmentation, as also affirmed by this reviewer.
2. Our AI system consistently outperforms these competing methods in all three experiments, especially for the case when using small training set (i.e., 100 scans). These

results show the advance of various strategies we proposed. For example, instead of simply localizing each tooth by points or bounding boxes as used in these competing methods, our AI system learns hierarchical morphological representations (e.g., tooth skeleton, tooth boundary, and root apices) for teeth often with varying shapes, and thus can more effectively characterize each tooth even with blurring boundaries using small training dataset.

In summary, compared to the existing deep-learning-based tooth segmentation methods, our AI system has at least three aspects of advantage. **First**, our AI system is fully automatic, while most existing methods need human intervention (i.e., having to manually delineate foreground dental ROI) before tooth segmentation. **Second**, our AI system has the best tooth segmentation accuracy due to the use of our proposed hierarchical morphological representation. **Third**, to the best of our knowledge, our AI system is the first deep-learning work for joint tooth instance and alveolar bone segmentation from CBCT images, thus largely improving computational efficiency.

Table R1. Segmentation results of different methods trained on limited or large dataset, and tested on the external dataset with many abnormalities.			
Methods	Training set (cases)	Dice (Tooth) (%)	Dice (bone) (%)
ToothNet* [1]	100	84.9	-
	100 ⁺	85.4	-
	3172	90.6	-
MWTNet* [2]	100	83.7	-
	100 ⁺	85.6	-
	3172	90.4	-
CGDNet* [3]	100	85.8	-
	100 ⁺	87.2	-
	3172	91.0	-
Ours	100	87.6	88.0
	100 ⁺	88.3	88.9
	3172	92.5	93.8

“*” means the method needs manual intervention (i.e., manually delineate foreground dental ROI). “+” means data augmentation techniques are used, including image flip, rotation, random deformation, and the learning based generative model.

We have added the comparison in **Result** section to demonstrate the advantage of our method.

Comment #2: The authors' comments about previous works are misleading. The existing methods are also fully automatic and can deal with metal artifacts and missing teeth. Also, there is no reason to believe that the proposed method is effective for handling metal artifacts.

Response: Thanks for this great review comment, for our possibly unclear description. We agree with the reviewer that, given a predefined region of interest (ROI), most existing methods can segment teeth automatically. But, ROIs often have to be located manually in the existing methods (i.e., ToothNet [1] and CGDNet [3]), thus, in this sense, the whole process for teeth segmentation from original CBCT images is not fully automatic.

We also agree that some existing methods have attempted to address the challenging cases with metal artifacts and missing teeth. For example, a dense ASPP module has been designed in CGDNet [3] for this purpose, with state-of-the-art segmentation accuracy (but on a very small dataset with only 8 CBCT scans). It is worth noting that our AI system can more robustly handle the challenging cases (including metal artifacts) than CGDNet, as demonstrated by new experimental results in Table R1, using either limited dataset or large-scale dataset. This is mainly due to the use of our two proposed complementary strategies for explicitly enhancing the learning of geometric tooth shapes in the CBCT images (even with metal artifacts or blurry boundaries). 1) We explicitly capture tooth skeleton information to provide rich geometric guidance for the downstream tooth segmentation. 2) We use tooth boundary prediction as an auxiliary task for tooth segmentation, thus explicitly enhancing the network learning at tooth boundaries even with limited intensity contrast (i.e., in the case of metal artifacts).

We have included all above clarifications in **Discussion** section of our revised paper to clearly compare our method with existing methods.

Comment #3: This paper lacks a detailed analysis of why the proposed method is better.

Response: Based on this great comment, we make a comprehensive revision of our paper by including new experiments, ablation studies, and associated discussions (as included in Results and Discussion sections of the revised paper) to systematically justify why our AI system is better.

Overall, the three main technical contributions make our AI system better than the existing methods:

1. Concurrent segmentation of tooth instances and alveolar bones in a fully automatic fashion, which is the first work in this study, to our best knowledge. In this way, the whole pipeline in our AI system can be tested without manual intervention, including the dental ROI localization, tooth segmentation, and alveolar bone segmentation from the original CBCT images.

2. Explicit learning of hierarchical morphological representation for accurate segmentation of teeth even with challenging appearances (i.e., with metal artifacts or missing teeth). Specifically, our method can automatically identify the tooth centroid and skeleton for better localizing teeth and capturing their complicated shapes, than using the simple bounding-box representation in other existing methods. Moreover, our method performs tooth segmentation in a multi-task learning setting, by capturing the intrinsic relatedness from multi-level geometric perspectives (i.e., tooth boundaries, apices, and masks) to boost the training of the segmentation network. Due to the use of such a hierarchical morphological representation, our method can produce more accurate tooth segmentation results, as already illustrated in our responses to Comments #1 and #2.
3. Enhancing intensity contrast between alveolar bones and soft tissues for more accurate delineation of in-between boundaries and thus better segmentation of alveolar bones, by using a filter-enhanced (i.e., Harr transform) cascaded network.

In addition to direct comparison of our AI system with other existing methods (i.e., shown in Table R1), we have also included a set of ablation studies for more carefully evaluating the efficacy of each important component in our AI system, including the skeleton representation, multi-task learning scheme, and Harr filter for bone segmentation. The results are shown in Table R2. As for the tooth segmentation, we train three competing models, i.e., 1) our AI system (**AI**), 2) our AI system without skeleton information (**AI (w/o S)**), and 3) our AI system without the multi-task learning scheme (**AI (w/o M)**). It can be found that AI (w/o S) and AI (w/o M) show relatively lower performance than AI in terms of all metrics (e.g., Dice score of 2.3% and 1.4% on internal set, and 1.4% and 1.1% on external set), demonstrating the effectiveness of our hierarchical morphological representation for accurate tooth segmentation. Moreover, the multi-task learning scheme with boundary prediction can greatly reduce the average surface distance error, especially on the CBCT images with blurry boundaries (e.g., metal artifacts). Next, as for the alveolar bone segmentation, we compare our AI system with the model without harr filter enhancement (**AI (w/o H)**). We find that our AI system increases the Dice score by 2.7% on internal testing set, and 2.6% on external testing set, respectively. The improvements are significant, indicating that enhancing intensity contrast between alveolar bones and soft tissues is helpful for bone segmentation network to learn more accurate boundaries.

We have accordingly revise our paper by adding new experiments, ablation studies, and associated discussions (as included in **Results** and **Discussion** sections of the revised paper) to systematically justify why our AI system is better.

Class	Model	Internal testing set			External testing set		
		Dice (%)	Sen (%)	ASD (mm)	Dice (%)	Sen (%)	ASD (mm)
Tooth	Ours (w/o S)	91.8	91.6	0.19	91.1	89.9	0.24
	Ours (w/o M)	92.7	91.9	0.23	91.4	90.3	0.29
	Ours	94.1	93.9	0.17	92.5	92.1	0.21
Bone	Ours (w/o H)	91.8	89.4	0.54	90.2	88.3	0.67
	Ours	94.5	93.8	0.33	93.8	93.5	0.40

Ours (*w/o* S): our AI system without skeleton information in the inputs of the 2nd stage tooth segmentation network.
Ours (*w/o* M): our AI system without the multi-task learning scheme, and only predict the tooth mask of the 2nd stage tooth segmentation network.
Ours (*w/o* H): our AI system without the harr filter enhancement for bone segmentation

Comment #4: Dental CBCT image quality depends on factors (e.g., tube voltages and tube current) that influence patient radiation dose. This paper mentions only hospitals that obtain CBCT data, and does not provide information on the types of CBCT machines used and radiation dose (tube current, tube voltage, etc.).

Response to Comment #4: Based on this constructive suggestion, we have provided the corresponding the CBCT machines, and radiation dose information (i.e., tube current and tube voltage) in Table R2 for more detailed description of the large-scale dataset.

Cohorts		Internal set			External testing set
		CQ-Hospital	HZ-Hospital	SH-Hospital	
Imaging protocols	Manufacturer	Imaging Sciences International	Planmeca	Vatech; Sirona	Instrumentarium Dental; LargeV; Bondent Imaging; Carestream Health; Trophy; FUSSEN; PointNix; HDXWILL; GENORAY; SOREDEX; RAY; MEYER;
	Manufacturer's Model	17-19	ProMax	PHT-6500; XG3D	OP300; HighRes3D; Bondream 3D-1020; CS 9300; K9500; Point Combi 500; DENTRI-C; PAPAYA 3D; Cranex3D; RAYSCAN; SS-X10010DPlus; SS-X9010DPro; SS-X9010DPro-3DE
	Tube voltage (kvp)	80-120	80-90	70-100	70-120
	Tube current (mA)	5-12	4-14	3-10	4-13

We have accordingly added the corresponding the CBCT machines, and radiation dose information (i.e., tube current and tube voltage) in **Table 1** of the manuscript.

Comment #5: It seems that the only contribution of this paper is that the amount of data is very large.

Response to Comment #5: Thank the reviewer for agreeing our contribution of collecting a large-scale dataset of real-clinical dental CBCT images. However, we'd like to clarify that this work also has two other equally important contributions:

1. *Technically*, we proposed a fully automatic deep-learning approach for accurate tooth instance and bone segmentation from CBCT images, which has been well described in our responses to the reviewer's other comments.
2. *Clinically*, with the large dataset and accurate segmentation results, our AI system offers great potential to improve clinical decisions and treatments in digital dentistry. For example, the clinical findings derived from this study (e.g., tooth volume and density changes across different ages) may be a basis for future research in the dental society, which has also been affirmed by Reviewer 4 from a clinician's perspective.

We have accordingly revised the manuscript in **Discussion** to clarify the contribution of our work more clearly.

To Reviewer #2:

This paper proposes an automated tooth segmentation method in CBCT images. It first extracts the ROI. Next, it detects skeleton and then predicts tooth apices and boundary. The tooth region is segmented by faster clustering using the centroid. First of all, I would like to congratulate you on your excellent performance.

Summary of Responses below: We'd like to thank the reviewer for supporting and insightful suggestions. Accordingly, we have thoroughly revised the submitted manuscript to improve its quality. The response to each comment is provided as follows.

Comment #1: Does it recognize tooth numbers? Maybe no. I am a little confused because the experimental results show the comparison.

Response to Comment #1: Thank the reviewer for pointing out the unclear description. Our AI system can automatically recognize the number of teeth under segmentation and identify the corresponding tooth ID based on ISO notation [1].

Specifically, given a dental ROI extracted from CBCT images, the tooth centroid network can generate a heatmap where each tooth centroid region has a peak value. Following a faster clustering operated in the centroid heatmap, we can detect each tooth object and simultaneously recognize tooth numbers. Then, as shown in **Fig. R1**, the encoder part of the single tooth segmentation network is followed by a max-pooling layer and three fully-connected layers to identify the category of the input CBCT patch (cropped at the tooth location).

We have accordingly revised the manuscript in **Method** to describe this point more clearly.

Fig. R1. The detailed network architecture of the single tooth segmentation network with multi-task scheme.

Comment #2: Show the data distribution of the abnormalities in the training and testing dataset.

Response to Comment #2: Following this constructive suggestion, we have added the data distribution of the abnormalities in the training and testing dataset. Generally, the typical abnormality types include metal artifacts, missing teeth, and crowded teeth. As shown in **Table. R3**, we can find the internal testing set and the training set have similar distributions of dental abnormalities, as they are randomly sampled from the same large-scale data pool. In contrast, since the external testing set is collected from different dental clinics, the distribution of its dental abnormalities is a little different compared with the internal set. As the experimental results summarized in Table. 2 in the revised manuscript, our AI system consistently obtains the state-of-the-art performance on these heterogeneous testing sets, implying its generalizability in clinical practice. Notably, some subjects may simultaneously have more than one abnormality.

Table R4. The data distribution of the abnormalities in the training and testing dataset.			
Dental abnormalities	Internal set		External set
	Training dataset (3172 CBCT scans)	Internal testing dataset (1359 CBCT scans)	External testing dataset (407 CBCT scans)
Missing teeth	946	345	137
Misalignment	2351	1085	314
Metal artifacts	435	199	96

As suggested, we have accordingly added Table 2 in the revised manuscript, and clearly stated the data distribution in **results** section.

Comment #3: Discuss the reason why this method achieves better results for abnormalities.

Response to Comment #3: Thanks for the great comment. In the revision, we have included associated discussions with new ablation studies to clarify why our method can handle dental abnormalities more effectively than other existing methods.

The main reason is the targeted methodological designs. Our AI system explicitly learns hierarchical morphological representations for precise tooth instance delineation from CBCT images with challenging appearances (e.g., missing teeth or metal artifacts). Specifically, our method can automatically identify the tooth centroid and skeleton, which are more stable descriptors in localizing teeth and capturing their complicated shapes, compared to the bounding-box representations used in other existing approaches. In addition, our method performs the individual tooth segmentation in a multi-task learning scheme, where the intrinsic relatedness from multi-level geometric perspectives (i.e., tooth boundaries, apices, and masks) is captured to boost the learning of a segmentation network, especially at the tooth boundary/root areas with blurred signals.

In addition, we have collected the largest dataset so far (4,215 patients with 4,938 CBCT images) to train our AI system. More representative features can be extracted for robust prediction even in challenging cases with varying dental abnormalities.

We have accordingly added more details in **Comparison** and **Discussion** of the manuscript to describe why this method achieves better results for abnormalities.

To reviewer #3:

This paper presented a deep learning-based system for automatic tooth and alveolar bone segmentation from CBCTs. The authors have collected a large CBCT dataset from 4215 patients (with 4938 CBCT scans) of 15 different centers. The proposed model reports a performance gain against experienced radiologists in terms of the average Dice similarity coefficient. The experimental results are plausible.

Summary of Responses: We'd like to thank the reviewer for the careful review of our paper. We also appreciate the encouraging and constructive comments. According to review comments, we have carefully revised the manuscript by adding more detailed description of our proposed method and analyzing its effectiveness more comprehensively. The response to each specific comment is provided below.

Comment #1: The proposed model was trained and validated on a large dataset. How about the performance gains compared with the existing deep learning-based tooth segmentation model learned from limited data? The data augmentation techniques have been widely used in medical image segmentation, where the generative model is effective in generating a large dataset with annotations [a]. It would be helpful to discuss the advantages of large data.

[a] Zhao, Amy, Guha Balakrishnan, Frédo Durand, John V. Guttag and Adrian V. Dalca. "Data Augmentation Using Learned Transformations for One-Shot Medical Image Segmentation." 2019 IEEE/CVF Conference on Computer Vision and Pattern Recognition (CVPR) (2019): 8535-8545.

Response to Comment #1: Following this constructive suggestion, we have added new experiments to directly compare our AI system with several most representative deep-learning-based tooth segmentation methods (i.e., **ToothNet** [1], **MWTNet** [2], and **CGDNet** [3]). Specifically, to have a fair and comprehensive validation, our method was compared with these state-of-the-art approaches in three different scenarios, including 1) a small-sized training dataset (100 CBCT scans), 2) a small-sized training dataset with data augmentation techniques (100+ CBCT scans), and 3) a large-scale dataset (3,172 CBCT scans). Note that the reference [a] indicated by the reviewer increase brain MRI data for tissue segmentation by leveraging predefined atlas (i.e., a template image). However, since no human tooth atlas exists in the literature, we choose another more general data augmentation method based on Conditional-GAN [4] in this study.

The corresponding results are presented in Table R1. From Table R1, we can have two important observations:

1. Compared with the models trained on a small-sized set (i.e., 100 CBCT scans), the data augmentation techniques (100+ CBCT scans) can consistently improve the segmentation accuracy for all competing methods, although the improvement is not significant.

2. Compared with the models trained on a small-sized set with data augmentation techniques (100+ CBCT scans), the use of real-clinical large-scale data (3,172 CBCT scans) leads to more significant performance gains for all competing methods (including our AI system).

These observations indicate that data augmentation, including traditional (i.e., random flip, rotation, and deformation) and learning-based (i.e., Conditional-GAN) techniques, indeed can boost the performance of a deep network on a small-sized training dataset. However, the improvements are limited compared to the large-scale dataset collected from real-world clinics. It is mainly because such a small-sized real dataset, as well as the synthesized data augmented from them, cannot completely cover the dramatically varying image styles and data distributions caused by different image protocols, scanner brands, or parameters in clinical practice. The results shown in Table 3 further demonstrate the essentialness of collecting a large-scale real-clinical dataset for segmentation model development.

In addition, we also find that our AI system consistently outperforms the existing deep learning-based methods in all cases (especially when the training set is limited), demonstrating the effectiveness of the proposed hierarchical tooth topological representation for tooth segmentation.

We have accordingly added a **Comparison**, and revised the **Discussion** to discuss the advantage of large data.

Comment #2: What is the image styles and data distributions variations across different centers and manufacturers? It is not clear on the capacity of the proposed system on CBCTs of different styles and distributions.

Response to Comment #2: Thanks for the great suggestion, we have accordingly revised the manuscript to describe the heterogenous multi-center data more clearly.

First, we have added **Table R2** to summarize the scanner manufacturers and imaging protocols (e.g., radiation dose tube current, and tube voltage) in different centers. It can be seen that the large-scale dataset is collected from different manufactures with diverse imaging protocols, which usually has a large difference in image appearances.

Second, we have included **Fig. R2** to show the heterogeneous intensity histograms of the CBCT data collected from different centers and manufacturers, including the internal set of CQ-Hospital (Imaging Sciences International|17-19), HZ-Hospital (Planmeca|ProMax), and SH-Hospital (Vatech|PHT-6500, Sirona|XG3D), and four typical manufacturers of the external set (Instrumentarium Dental|OP300, PointNix|Point 800 3D Plus 8N, HDXWILL|DENTRI-C, and GENORAY|PAPAYA 3D). From Fig. R2, we can find that there are large data variations across different centers and manufacturers. Thus, it is essential to collect a large-scale dataset to develop an AI system with promising generalizability and robustness.

Fig. R2. The overall intensity histogram distributions of the CBCT data collected from different manufacturers.

We have accordingly revised the manuscript in the **Study design and participants** part of **Section Results** to describe the heterogenous multi-center data more clearly.

Comment #3: The ground truth annotations were obtained by the manual annotation of senior radiologists with ten or more years of professional experience. The manual annotation is time-consuming and prone to inter-practitioner’s variations. How about the time complexity of the ground truth annotation and the scheme to avoid inter-practitioner’s variations regarding the hierarchical data annotation method?

Response to Comment #3: Thanks for the suggestion. We have accordingly revised the supplementary materials to describe the time complexity of the ground truth annotation.

On average, it takes about 2-3 hours to manually label one CBCT scan, and 5-10 minutes to refine one CBCT scan based on the predictions from our AI system. Generally, the 1st stage of the fully manual annotation process (100 CBCT scans independently annotated by 3 senior radiologists) takes about 2 months. The 2nd (600 CBCT scans refined by 10 junior radiologists and double-checked by 3 senior radiologists independently) and 3rd (4238 CBCT scans refined by 10 junior radiologists and double-checked by 3 senior radiologists independently) stages of label refinement processes take about 5 days and 40 days, respectively. Overall, we have spent about 4 months annotating the dataset with 3 senior radiologists and 10 junior radiologists.

In the data annotation process, to avoid inter-practitioner's variations, all CBCT scans are labeled and checked by the 3 senior radiologists. When disagreements occurred among the 3 senior radiologists, they will have a discussion to make the final decision.

We have accordingly revised the supplementary materials to describe the time complexity of the ground truth annotation.

Comment #4: As shown in Fig. 7, the output mask of the Cascaded bone segmentation network includes the tooth. It is unclear whether the mask of alveolar bones was obtained directly from the end-to-end model or by some postprocessing, e.g., Boolean operation with the tooth segmentation.

Response to Comment #4: Thanks for pointing out the mistake. Actually, the cascaded bone segmentation network only predicts the bone mask, and the tooth label is not included. To merge the predicted bone and tooth masks as the final results, a postprocessing step of label voting is also needed. For example, if one voxel is simultaneously predicted as bone and tooth, we will compare the probabilities predicted by the bone and tooth segmentation networks, and choose the label with a larger probability as the final prediction.

We have accordingly revised the manuscript in the Method description and Fig. 7 to describe the output mask more clearly.

Comment #5: The proposed system was used to validate the automation and clinical applicability. Fig. 6. Illustrates the curves of tooth volumes and intensities over different patient ages regarding different types of teeth. How about the difference between the curves computed from the automatic segmentation and manual annotation?

Response to Comment #5: As suggested, we have included the changing curves of tooth volumes and intensities computed from the automatic segmentation, and compared them with those derived from manual annotations. The corresponding results are presented in Fig. R3 in this reply letter.

By comparing Fig. R3(a) with (b), we can find that the trajectories are very similar, indicating that the manual annotation and automatic segmentation produced by our AI system can yield the same clinical findings.

The reason why this paper uses manual annotation to compute the tooth volume and intensity changing curves is that we aim to explore clinically meaningful and exciting findings from the large-scale CBCT dataset. Thus, the statistical results computed from ground truth annotation will be more convincing.

Fig. R3. The changing curves of the tooth volumes and intensities over different patient ages. (a) The changing curves computed from manual annotation. (b) The changing curves computed from automatic segmentation.

We have accordingly added the description that why we choose manual annotation, instead of automatic segmentation, to explore the clinical knowledge in **discussion**.

Comment #6: The Harr transform was used for image enhancements for bone segmentation. Why is such transform not used for tooth segmentation? Since the learned convolutional kernels were used as image filters in feature extraction, it is unclear whether the Harr transform-like preprocessing is necessary for the image segmentation framework.

Response to Comment #6: Thanks for this great comment. In this paper, the Harr transform is used to enhance the image intensity contrast between bones and soft tissues, which is helpful for the alveolar bone segmentation network to learn more accurate boundaries. For example, when compared with the network only inputting the original CBCT images, the inclusion of boundary-enhanced image (produced by the Harr transform) effectively improved the bone segmentation accuracy from 91.8% to 94.5% on the internal testing set (shown in Table R2).

On the other hand, we should indicate that such a simple boundary enhancement strategy is not helpful for tooth instance segmentation, mainly due to the task property. That is, the specific challenges in tooth segmentation are the accurate delineations of the boundaries

between neighboring teeth and those between tooth roots and surrounding alveolar bones. Unfortunately, it is difficult for the simple Harr operation to effectively address the issue of very similar intensities between different targets. Our experiments also show that results of tooth segmentation are comparable (94.0% v.s., 94.1%) with or without harr transformation.

We agree with the reviewer that the learned convolutional kernels in deep CNNs can be regarded as image filters for feature extraction. In this study, Harr transformation is leveraged as featured prior knowledge (i.e., enhanced boundaries or edges from various filtering operations), which can explicitly and effectively guide the network learning of alveolar bone segmentation. Such combinations of data-driven and knowledge-driven approaches have demonstrated promising performance in many learning tasks, including tissue segmentation [5], image generation [6], or depth estimation [7].

We have accordingly added the **ablation study** in **results** section, and discuss the advantage of Harr transform in **discussion**.

Comment #7: What is the input of the multi-task tooth segmentation network? As shown in Fig. 7, the tooth skeleton and centroid served as the input of the segmentation network. Whether the ROI features was used in tooth segmentation?

Response to Comment #7: Thanks for pointing out the mistake. Actually, there are three inputs of the multi-task tooth segmentation network, including the patches cropped from the tooth centroid map, the skeleton map, and the tooth ROI images, respectively. We use the original ROI images instead of the ROI features in the multi-task tooth segmentation network, mainly due to the GPU memory costs. Considering the large volumes of ROI area (256x256x256) and tooth patch (96x96x96), we have to train the tooth centroid/skeleton prediction networks and the single tooth segmentation network separately, instead of using ROI features as an end-to-end training scheme.

Following the comment, we have accordingly updated the **method** section and Fig. 7 of the manuscript to describe this point more clearly.

Comment #8: The proposed system performed the segmentation of teeth and alveolar bones better than expert radiologists. Does the slight performance gain have statistical significance? How about an annotation difference between the system and experts regarding the slight improvements?

Response to Comment #8: Thanks for the insightful comment. Accordingly, we have conducted a paired t-test evaluation to validate whether the improvements by our AI system have statistical significance. The significance level is set as 0.05. Specifically, for tooth segmentation, the paired p-values are $2e-5$ and $7e-3$ when compared our method with expert-1 and expert-2, respectively. And for alveolar bone segmentation, the paired p-values are $1e-3$ (expert-1) and $9e-3$ (expert-2).

The results show that all the p-values are much smaller than 0.05, indicating that the improvements over manual annotation are statistically significant.

Qualitatively, our AI system also outperforms the experts in two-fold. *First*, the 3D surfaces of teeth and bones reconstructed by our AI system are much smoother compared with the manual annotation, such as the typical examples shown in Fig. 5 of the manuscript. *Second*, with the advantage of conducting segmentation in 3D space (i.e., 3D CNNs), our system can more efficiently annotate the touching interface between the upper and lower teeth under a close bite condition.

We have added more details about the clinical validation and accordingly revised the manuscript in **results** section.

To reviewer #4:

Please note that my comments are from a clinician's perspective. In light of that, this manuscript describes an exciting clinical complement. The group earns applause for creating a useful, validated tool for clinicians as well as for engaging a multidisciplinary team in the system's development.

Summary: All in all, this was a great paper to read. The clinical relevance is good and offers great potential to improve clinical decisions as well as to improve the ability to assess treatment outcomes qualitatively and quantitatively over time. I look forward to seeing a similar report in the dental literature.

Summary of Responses: We would like to thank the reviewer for the support, especially from a clinician's perspective to validate our work's exciting clinical value. Following the insightful comments and constructive suggestions from the reviewer, we have carefully revised the manuscript to present the details of our method more clearly. The point-to-point responses are summarized below.

Comment #1: The inclusion of Figure 6 showing volume and density changes by tooth across different ages is interesting and might be the basis for another publication, perhaps in the dental literature. It would be interesting to have the authors conjecture why there is a peak in the volume trajectory curve for middle-aged patients.

Response to Comment #1: Thanks for the reviewer's interest in the findings of volume and density changes across different ages. To explore why there is a peak in the volume trajectory curve for middle-aged patients, we have an in-depth discussion with many professional dentists, and check the large-scale dataset. The potential reasons could be two-fold.

1. *First*, there is a significant tooth size discrepancy across people from different regions [8][9]. And in this study, our dataset (i.e., internal and external sets) is mainly collected from three places (i.e., Chongqing, Hangzhou, and Shanghai), where their tooth size distributions may be slightly different, and lead to the peak in the volume trajectory curve for middle-aged patients.
2. *Second*, all the CBCT images are collected from patients seeking different dental treatments in hospitals, which may also produce peak value in the volume trajectory curve. For example, in clinics, the volumes of caries usually become slightly larger after restoration treatments, especially measured on CBCT images where the restorative materials (e.g., metal or ceramic) lead to higher intensity values and larger tooth size.

Hence, in the future, we plan to collect larger data from more centers, and calculate the tooth volume and intensity trajectories with different scenarios, including inter- and intra-different regions, and before and after dental treatments.

We have accordingly revised the manuscript in **discussion** to provide more details.

Comment #2: It might be useful in future publications describing the system to address the diversity of the population that formed the basis for analysis. For instance, are all of the patients from a single culture, is a breadth of minorities included in the data set, etc.

Response to Comment #2: Thanks for the great suggestion. Currently, our large-scale CBCT data are mainly collected from three places, including Chongqing, Hangzhou, and Shanghai. Their cultures (e.g., eating habits) are a little different. For example, Chongqing is the spicy food center of China, and more patients in CQ-Hospital like eating spicy cuisine compared to other centers. But a breadth of minorities is not included in the dataset at this stage, and most people are of Han nationality.

In the future, we plan to collect more data from multinational different centers to improve the diversity of the population.

We have listed this point as a future research direction in the conclusion part of our manuscript.

Comment #3: It might be useful to somewhere add the definition of “DICE” for those clinicians who are not familiar with the term and the significance of the values that your report.

Response to Comment #3: As suggested, we have revised the manuscript for a more clear description of “DICE” and other metrics used in this study for segmentation performance quantification.

Comment #4: Throughout the document there is inconsistent use of () around e.g., and i.e.

Response to Comment #4: Thanks for the detailed review. These errors have been corrected accordingly.

Comment #5: The last lines of the introduction section address numbers of images used to validate the AI system. The numbers there differ from those used in the abstract and may be confusing, without further clarification, to the readers. I suggest that you reword those sentences to capitalize (and remind the readers) of the 4,215 patient records and nearly 5,000 CBCT scans you had access to and used.

Response to Comment #5: Thanks for the suggestion. We have updated the description accordingly.

Comment #6: More Specific comments:

- Line 33: perhaps the word 'speedup' could be replaced by 'faster'
- Line 40: generally, in clinical use, the term dental implant is used instead of dental implanting.
- Line 43: generally, in clinical use, the line might read "...X-rays, 3D Intra-Oral scans, and ...(CBCT) images."
- Line 62: replace 'changing appearances' to variations in structures'
- Line 64: change 'small-sized data' to 'small-sized data sets'
- Line 83: change 'implanting' to 'implants' and 'we collected 4,215' to 'we collected data of 4,215'
- Line 93: change 'CBCT' to 'of the CBCT'
- Line 107: 'Method' should be 'Methods'
- Line 124: Perhaps 'segment' would be a better word than 'extract'
- Line 142: Here you state that the radiologists had more than 5 years' experience. Elsewhere in the document you state that they had 10 years' experience. Are they different radiologists? Perhaps some clarification would be useful to avoid confusion.
- Line 171: You point out that 12 of 100 CBCT scans required extra-human intervention. It would be useful to know why that intervention was necessary.
- Line 178: 'Discussions' should be 'Discussion'
- Line 219: Perhaps better to replace 'become firmer' with 'changes'.

Response to Comment #5: Thanks for the careful suggestion. For these specific comments, we have accordingly corrected our manuscript.

Reference:

- [1] Cui et al. "ToothNet: automatic tooth instance segmentation and identification from cone beam CT images." *Proceedings of the IEEE/CVF Conference on Computer Vision and Pattern Recognition*. 2019.
- [2] Chen et al. "Automatic segmentation of individual tooth in dental CBCT images from tooth surface map by a multi- task FCN." *IEEE Access* 8 (2020): 97296-97309.
- [3] Wu et al. "Center-sensitive and boundary-aware tooth instance segmentation and classification from cone-beam ct." *2020 IEEE 17th International Symposium on Biomedical Imaging (ISBI)*. IEEE, 2020.
- [4] Mirza, Mehdi, and Simon Osindero. "Conditional generative adversarial nets." *arXiv preprint arXiv:1411.1784* (2014).
- [5] Lian, C., Zhang, J., Liu, M., Zong, X., Hung, S. C., Lin, W., & Shen, D. (2018). Multi-channel multi-scale fully convolutional network for 3D perivascular spaces segmentation in 7T MR images. *Medical image analysis*, 46, 106- 117.
- [6] Nestmeyer, Thomas, and Peter V. Gehler. "Reflectance adaptive filtering improves intrinsic image estimation." *Proceedings of the IEEE Conference on Computer Vision and Pattern Recognition*. 2017.
- [7] Qi, X., Liu, Z., Liao, R., Torr, P. H., Urtasun, R., & Jia, J. (2020). Geonet++: Iterative geometric neural network with edge-aware refinement for joint depth and surface normal estimation. *IEEE Transactions on Pattern Analysis and Machine Intelligence*, 2020.
- [8] Bishara, S. E., Jakobsen, J. R., Abdallah, E. M., & Garcia, A. F. (1989). Comparisons of mesiodistal and buccolingual crown dimensions of the permanent teeth in three populations from Egypt, Mexico, and the United States. *American Journal of Orthodontics and Dentofacial Orthopedics*, 96(5), 416-422.
- [9] DeVaughan, T. C. (2017). Tooth Size Comparison Between Citizens of the Chickasaw Nation and Caucasians.

Acknowledgement:

We would like to thank Dr. Min Gu for variable discussion in clinical perspective, especially in the tooth volume and intensity trajectory across different ages.

Reviewers' Comments:

Reviewer #1:

Remarks to the Author:

The authors have adequately revised their manuscript and publication is recommended.

Reviewer #2:

Remarks to the Author:

This paper proposes an automated method for segmenting teeth region and identifying teeth number in CBCT images. The proposed method outperforms the conventional methods including ToothNet, MWTNet, and CGDNet. The authors revised well according to reviewers' comments.

Reviewer #3:

Remarks to the Author:

I am glad that the authors have addressed most of the comments. I still have some minor concerns.

* The authors provide intensity histogram distributions of CBCTs from different centers. There lack of discussions on the dentition shape or morphology distributions from different centers. I think the coverage of a variety of tooth and bone shapes, especially of patients with dental problems, by such a large dataset with 4938 CBCTs would be helpful to explain the performance gap in Table r1.

* I agree that integrating data-driven and knowledge-driven approaches would be helpful in particular tasks. Though, the referred work in the reply, such as [6], did not utilize the Harr transform-like processing.

Point-to-point Responses to Review Comments

Manuscript ID: NCOMMS-21-40797R

Paper Title: A fully automatic, clinically applicable AI system for tooth and alveolar bone segmentation from cone-beam CT images

Dear Editor and Reviewers,

Thank you for encouraging feedbacks on our previous responses. We have now revised our paper according to new suggestions from R3, the revised parts indicated in blue font. The point-to-point response to each comment, along with our respective revision in our paper, are provided below.

To Reviewer #3:

I am glad that the authors have addressed most of the comments. I still have some minor concerns.

Summary of Responses below: We thank the reviewer for positive feedback on our previous revision. Following new constructive suggestions, we have carefully revised our paper accordingly, with the point-to-point responses presented below.

Comment #1: The authors provide intensity histogram distributions of CBCTs from different centers. There lack of discussions on the dentition shape or morphology distributions from different centers. I think the coverage of a variety of tooth and bone shapes, especially of patients with dental problems, by such a large dataset with 4938 CBCTs would be helpful to explain the performance gap in Table r1.

Response: Thanks for insightful suggestion. We agree with the reviewer that the coverage of a variety of tooth and bone shapes across different centers, especially from patients with dental problems, is also important to improve the robustness and generalizability of our AI system.

The large-scale, multi-center CBCT imaging data studied in this paper do present large variations in terms of dentition shape/morphology, which is described in detail two facts below:

- 1) These large set of CBCT images (i.e., 4,938 CBCT scans of 4,215 patients) were acquired from 3 hospitals (in Chongqing, Hangzhou, and Shanghai) and 12 dental clinics (widely distributed over different regions of China). As reported by the oral health survey [1][2], the dentition distributions could be different across people from different regions.
- 2) These CBCT images were scanned from patients with varying dental conditions. For example, most patients in hospital of Chongqing suffered from the problem of missing teeth with the alveolar bone resorption, while most patients in hospital of Shanghai received orthodontic treatments. Such differences further lead to large variations of dentition shape and/or morphology distributions across different centers.

As supported by experimental results in Table r1 of our previous response letter, such a large-scale and heterogeneous dataset led to a much more robust and generalizable AI system for both tooth and bone segmentations.

Following this suggestion, we have accordingly updated the Discussion section of our paper to discuss in more detail the significance of collecting a large-scale dataset with heterogeneous distributions of dentition shape and/or morphology across different centers.

Comment #2: I agree that integrating data-driven and knowledge-driven approaches would be helpful in particular tasks. Though, the referred work in the reply, such as [6], did not utilize the Harr transform-like processing.

Response: Thanks for careful review and indicating the typo. We have now corrected the reference (as [3] provided below in this response letter).

Specifically, Fan *et al.* [3] propose a deep learning network for intrinsic image decompositions by leveraging the edge map to highlight key sparse structures. Lian *et al.* [4] use an additional image processed by the harr-transform filter to afford tubular structural information for perivascular space segmentation. And, Qi *et al.* [5] integrate Canny edge information into a neural network for joint depth and surface normal prediction from a single image. All these works suggest that specific designs to combine data-driven and knowledge-driven approaches would be helpful for medical image computing tasks.

According to this comment, we have accordingly revised the Discussion section of our manuscript to discuss in more detail the advantage of integrating data-driven and knowledge-driven methods.

Reference

- [1] Bishara, S. E., Jakobsen, J. R., Abdallah, E. M., & Garcia, A. F. (1989). Comparisons of mesiodistal and buccolingual crown dimensions of the permanent teeth in three populations from Egypt, Mexico, and the United States. *American Journal of Orthodontics and Dentofacial Orthopedics*, 96(5), 416-422.
- [2] DeV Vaughan, T. C. (2017). Tooth Size Comparison Between Citizens of the Chickasaw Nation and Caucasians.
- [3] Fan, Q., Yang, J., Hua, G., Chen, B., & Wipf, D. (2018). Revisiting deep intrinsic image decompositions. *In Proceedings of the IEEE conference on computer vision and pattern recognition* (pp. 8944-8952).
- [4] Lian, C., Zhang, J., Liu, M., Zong, X., Hung, S. C., Lin, W., & Shen, D. (2018). Multi-channel multi-scale fully convolutional network for 3D perivascular spaces segmentation in 7T MR images. *Medical image analysis*, 46, 106-117.
- [5] Qi, X., Liu, Z., Liao, R., Torr, P. H., Urtasun, R., & Jia, J. (2020). Geonet++: Iterative geometric neural network with edge-aware refinement for joint depth and surface normal estimation. *IEEE Transactions on Pattern Analysis and Machine Intelligence*, 2020.